# ISMIP-HOM benchmark experiments using Underworld

Till Sachau[1], Haibin Yang[2], Justin Lang[1], Paul Bons[1,3], Louis Moresi[2]

[1]Department of Geosciences, Eberhard Karls University Tübingen, Tübingen, Germany
[2]Research School of Earth Sciences, Australian National University, Canberra, Australia
[3]China University of Geosciences, Beijing, China

*Correspondence to*: Till Sachau (till.sachau@uni-tuebingen.de)

**Abstract.** Numerical models have become an indispensable tool for understanding and predicting the flow of ice sheets and glaciers. Here we present the full-Stokes software package Underworld to the glaciological community. The code is already well established in simulating complex geodynamic systems. Advantages for glaciology are that it provides a full-Stokes
solution for elasto-visco-plastic materials and includes mechanical anisotropy. Underworld uses a material point method to track the full history information of Lagrangian material points, of stratigraphic layers and of free surfaces. We show that Underworld successfully reproduces the results of other full-Stokes models for the benchmark experiments of the ISMIP-HOM project. Furthermore, we test FE meshes with different geometries, and highlight the need to be able to adapt the finite-element grid to discontinuous interfaces between materials with strongly different properties, such as the ice-bedrock
boundary.

## 1 Introduction

Numerical modeling has become a standard tool in the prediction of ice flow in ice sheets and glaciers and has gained increasing importance due to the quest to predict sea-level rise (Goelzer et al., 2017). Ice sheets and glaciers on Earth consist of ice 1h, the crystallographic variant of water ice that is stable under the conditions at the Earth's surface. Ice 1h is a
mineral with a hexagonal crystal symmetry that shows ductile or crystal-plastic behaviour (McConnell and Kidd, 1888; Nye, 1951; Glenn 1955; Budd and Jacka, 1989) at differential stresses in the order of 0.01-0.1 Mpa, that are typical for ice sheets.

The flow law of ice is generally assumed to be a power law (Glen, 1955; Budd and Jacka, 1989), often termed "Glen's (flow) law" (Haefeli, 1961), in which the strain rate is proportional to the differential stress to the power $n$, the stress exponent. Usually modelers assume $n=3$ (see e.g., Pattyn et al., 2008), although several studies – including the original study of Glen
(1955) – assume that $n \approx 4$ probably best describes the rheology of ice. This is confirmed by more recent studies (Goldsby and Kohlstedt, 2001; Goldsby, 2006; Bons et al., 2018; Ranganathan et al., 2021).

Most rock-forming minerals also flow with a power-law rheology (Ranalli, 1987; Evans and Kohlstedt, 1995). Modelers of tectonic processes thus face the same challenges related to non-linear flow as those in the glaciological community. Recent versions of the software package 'Underworld' (Moresi et al., 2007; Mansour et al., 2020a, 2020b, code available here:
https://doi.org/10.5281/zenodo.1436039) provide a Python API originally developed to simulate geodynamics processes. Similar to Elmer/Ice (Gagliardini et al. 2013), it solves the full Stokes equations for visco-elastic-plastic deformation and is coupled to heat flow (Moresi et al., 2003; Mansour et al., 2020). The latter is relevant considering the potential impact geothermal heat flow may have on ice flow and ice streams (Smith-Johnsen et al., 2020; Bons et al., 2021).

As with most minerals, the rheology of ice 1h strongly depends on a range of factors, such as temperature, microstructure,
etc. In addition, ice 1h has a strongly anisotropic rheology (Duval et al., 1983; Azuma, 1994) and it is increasingly recognized that that this plays a crucial role in the behavior of flowing ice, especially at ice streams (Rathmann and Lillien, 2022). Especially airborne radar (Schroeder et al., 2020) has shown a rich diversity in fold structures inside the ice sheets (NEEM community members, 2013; Wolovick, 2014; MacGregor et al., 2015; Bons et al., 2016; Cavitte et al., 2016; Leysinger-Vieli et al., 2018). Radar data also allow direct measurements of the crystallographic, and, hence, mechanical
anisotropy in ice (e.g., Young et al., 2021; Ershadi et al. 2022). As the mechanical anisotropy, together with processes such as basal melting, is thought to actively influence flow of ice and folding, there is an urgent need to include it in ice flow models on various scales (Rathmann and Lilien, 2022).

Underworld includes mechanical anisotropy (Moresi et al., 2006; Sharples et al., 2016). It employs the Material Point Method (MPM) (Sulsky et al., 1994; Moresi et al., 2003) where Lagrangian material points are combined with a Finite
Element mesh. First and foremost, these material points allow tracking of the strain history and rheological or physical changes on distinct Lagrangian points. Further, tracking of the material points allow us to understand the deformation of individual volumes or layers within the ice sheet and the evolution of the surface. Particles can also be used to record the

crystallographic preferred orientation (CPO), and thus the local mechanical anisotropy of the material. This way, the mechanical anisotropy can evolve as a result of the local deformation. The combination of both anisotropic rheology and particle tracking has potential for the modeling of large-scale folds of stratigraphic layers observed in ice sheets (Wolovick, 2014; NEEM community members, 2013; Bons et al., 2016; Cavitte et al., 2016; Leysinger-Vieli et al., 2018), in particular when the folding is a result of the anisotropic rheology of ice (Bons et al., 2016).

Finally, Underworld can be coupled with other models to investigate surface effects, such as sedimentation and erosion, and processes that affect the base of the model, such as mantle deformation and heat flux (Salles et al., 2016; Bahadori et al., 2022). These have their equivalents at the surface of ice sheets in the form of snow precipitation, ablation and both surface and basal melting (e.g. Jacobson and Raymond, 1998; Smith-Johnsen et al., 2020). For all these reasons Underworld, which is already well established for the simulation of complex tectonic processes (for instance Sandiford et al., 2020; Carluccio et al., 2019; Capitanio et al., 2019; Korchinski et al., 2018), surface processes (Bahadori et al., 2022) and long-term ground water motion (Mather et al., 2022), seems also well suited to simulate ice-sheet and glacier flow.

Any numerical model needs to be validated or benchmarked. The Ice Sheet Model Intercomparison Project for Higher-Order Models (ISMIP-HOM; Pattyn et al., 2008 and supplement or https://frank.pattyn.web.ulb.be/ismip/welcome.html) provides tests for the comparison of computational ice-sheet flow models for different purposes. "Higher-order" here refers to models that go beyond the shallow-ice approximation (SIA) up to full Stokes solutions (as Underworld does).

ISMIP-HOM includes both 2D and 3D experiments. The flow law is Glen's law with a stress exponent $n$=3 and, in one experiment, Newtonian flow. In this paper we publish the results for the full suite of experiments of the benchmark. We focus on three issues: (i) the viability of the results as compared to solutions provided by other models, (ii) the computation time, and (iii) the influence of the geometry of the underlying Finite Element grid. The tests are performed using the 2.10 release of the software package Underworld. Finally, we provide one example of how mechanical anisotropy and tracking of the stratigraphy can be incorporated in Underworld to illustrate the potential of Underworld to simulate mechanically complex systems and the resulting structures within a glacier or ice sheet.

## 2 Method

### 2.1 Governing equations

The solution in Underworld is based on the Stokes equation of slow flow of a Newtonian incompressible fluid:

$$\frac{\partial \sigma_{ij}}{\partial x_j} - \frac{\partial P}{\partial x_i} + \rho g_i = 0, \tag{1}$$

$$\frac{\partial v_i}{\partial x_i} = 0. \tag{2}$$

Here $\sigma_{ij}$ is the stress tensor, $P$ the pressure, $g$ the gravitational acceleration and $v$ the velocity (see Tables 1 and 2 for symbols used). Simulations are based on Glen's flow law for viscous flow (Glen, 1955), according to which the strain rate ($\dot{\epsilon}_{ij}$) is proportional to the deviatoric stress ($\tau_{ij}$) to the power $n$, the stress exponent. This flow law can be written as:

$$\dot{\epsilon}_{ij} = A \tau_{II}^{n-1} \tau_{ij}, \tag{3}$$

where $A$ is the temperature-dependent rate factor, $\tau_{II}$ the second invariant of the deviatoric stress tensor $\tau_{ij}$ (Nye, 1953).

Based on Newtonian flow, where $\tau_{ij} = 2\eta \dot{\epsilon}_{ij}$, we define an effective viscosity $\eta_{ice}$ after Eq. 3 as:

$$\eta_{ice} = \frac{1}{2} A^{-1/n} \dot{\epsilon}_{II}^{(1-n)/n}. \tag{4}$$

### 2.2 Parameters

| Symbol | Parameter | Value | Unit |
|---|---|---|---|
| $A_{n=3}$ | ice-flow parameter for stress exponent $n$=3 | $10^{-16}$ | $Pa^{-3}a^{-1}$ |
| $A_{n=1}$ | ice-flow parameter for stress exponent $n$=1 (Newtonian) | $2.140373 \times 10^7$ | $Pa^{-1}a^{-1}$ |

| $\rho_{ice}$ | ice density | 910 | kg m$^{-3}$ |
|---|---|---|---|
| $\rho_{bed}$ | bed rock density | 2700 | kg m$^{-3}$ |
| $n$ | exponent of Glen's flow law for ice | 3 or 1 | |
| $\eta_{bed}$ | constant bedrock viscosity | 1022 | Pa s |
| $\eta_{ice}$ | effective viscosity of ice | | Pa s |
| $g$ | gravitational constant | 9.81 | m s$^{-2}$ |
| $L$ | model width | 5-160 | km |

**Table 1: Parameters and their values, as prescribed by Pattyn et al. (2008) for the intercomparison project.**

| Symbol | Variable | Typical unit |
|---|---|---|
| $\beta^2$ | basal friction coefficient | |
| $\dot{\epsilon}_{ij}$ | strain rate tensor | a$^{-1}$ |
| $\dot{\epsilon}_{II}$ | second invariant of the strain rate tensor | a$^{-1}$ |
| $m$ | adaption parameter for mesh deformation | |
| $N$ | number of nodes in a mesh | |
| $P$ | pressure | Pa |
| $\boldsymbol{p}$ | coordinates of a vertex point | |
| $\tau_{xy}^{b}$ | horizontal shear stress at the ice basis in x-direction | Pa |
| $\tau_{ij}$ | deviatoric stress tensor | Pa |
| $\tau_{II}$ | second invariant of the deviatoric stress tensor | Pa |
| $v_x^b, v_y^b$ | velocity at ice basis, x- and y-component | m a$^{-1}$ |
| $v_x^s, v_y^s$ | velocity at ice surface, x- and y-component | m a$^{-1}$ |
| $x, y$ | axes parallel and vertical to the tilted surface, referred to as 'horizontal' / 'vertical' | m |

**Table 2: Symbols used in this paper and not listed in Table 1.**

### 2.2 Characteristics of Underworld with regard to specific challenges in the modeling of ice flow

85   Underworld is designed to solve some of the special problems relevant to modeling geodynamic processes. Identical problems arise in the modeling of ice. Some of these challenges are:

   a   the modeling of discontinuities in the material properties at layer boundaries, for instance at the ice-rock and ice-air interfaces;

   b   gradients within the ice, for instance due to strain softening or thermal effects;

90   c   the tracking of the strain history;

   d   the often-extreme spatial extent of the modeled systems;

   e   the very strong deformation of the material.

Underworld addresses these issues with the so-called material point method (MPM) (Sulsky et al., 1994; Moresi et al., 2003), which is closely related to the venerable particle-in-cell (PIC) method. MPM uses a Eulerian Finite Element mesh in
95   order to calculate the incremental development of the velocity field and other field variables, such as, for example, temperature and pressure. In the MPM method, Lagrangian material points ('particles') carry the density, viscosity, thermal conductivity, and other relevant material parameters. They thereby record the history at their current location at every time step and some historical properties like the stress at previous time step for simulations of viscoelastic deformations

(Farrington et al., 2014). The underlying mesh provides solutions for the incremental movement of the material points. The method is advantageous in the modeling of the emergence of structures (e. g. folding, see Mühlhaus et al., 2002) or where very strong deformation is involved, as in the deformation across shear zones or near the base of an ice sheet. In MPM, the mesh does not carry any history information other than deformation of the boundary and therefore can be re-meshed at any time as required and without loss of accuracy.

There is an unavoidable smoothing which comes from the coarseness of the computational mesh relative to the material point density (Moresi et al., 2003). While material boundaries are represented by a continuous interpolant on the grid, they are necessarily discrete in case of particles. This can lead to fluctuations in the solution close to sharp rheological or mechanical boundaries (Yang et al., 2020), for instance at the interface between ice and underlying rock. In the ice itself, a change in mechanical parameters is usually more gradual and is controlled, for example, by the temperature gradient.

Another complication in the numerical modeling of ice flow is the highly anisotropic behavior of ice, created by the near-orthotropic properties of the ice crystal. The possibility to model anisotropic flow is built into Underworld (Mühlhaus et al., 2004). Like any other local material property, the orientation of the anisotropy can be stored on the particle level.

Underworld offers a variety of possible solvers, including the well-known MUMPS, LU and multi-grid methods. We have carried out brief comparative precision tests with these solvers and could not find any difference in precision to the standard solver which is based on the multi-grid method. Throughout this study we will generally use MUMPS for 2D models and multi-grid for 3D models. The exception are tests of the computation time, for which we contrast a variety of solvers.

**3 Description of experiments**

The ISMIP-HOM benchmark experiments have been described in detail in Pattyn and Payne (2006) and Pattyn et al. (2008). We perform all the experiments as described in these publications. For simplicity we also apply the same alphabetical numbering scheme and refer to the experiments as Experiment 'A' to Experiment 'F'. As we strive to focus on the essentials in the following descriptions, we refer to the original publications for further technical details if needed.

Experiments 'A', 'B', 'C' and 'F' are three-dimensional. Experiments 'D' and 'E' consider only two spatial dimensions, and Experiment 'B' has an additional version in 2D. Experiments 'A' through 'D' are performed for a variety of horizontal system dimensions $L$, with $L$ = 5, 10, 20, 40 80 or 160 $km$. Experiment 'A' through 'E' use a flow law based on $n = 3$, experiment 'F' applies Newtonian flow where $n = 1$ (Table 1).

We tested the influence of the mesh geometry on the results and the CPU time consumption as a function of the total degree of freedom using Experiment 'D' and the 2D-version of Experiment 'B'.

**3.1 Experiments A and B**

A and B consider a slab of ice with a mean ice thickness $H = 1000$ m, lying on a sloping bed with a mean slope $\alpha = 0.5°$. The bedrock topography consists of a series of sinusoidal bumps (Experiment A) or ripples (Experiment B) with an amplitude of 500 m (Fig. 1). The minimum thickness of the rock layer is 500 m, and the total height of the model is 2000 m. The flow of ice is governed by Eq. (3). The bedrock viscosity is constant, and ice is frozen to the bedrock. Relevant material parameters are compiled in Table 1.

The surface elevation is described by the formula:

$$z_s(x, y) = - x \cdot \tan(\alpha), \tag{5}$$

bedrock topography for Experiment A is described using $z_s$ by:

$$z_b(x, y) = z_s(x, y) - 1000 + 500 \sin(\omega x) \cdot \sin(\omega y) \tag{6}$$

and for Experiment B by:

$$z_b(x, y) = z_s(x, y) - 1000 + 500 \sin(\omega x) \tag{7}$$

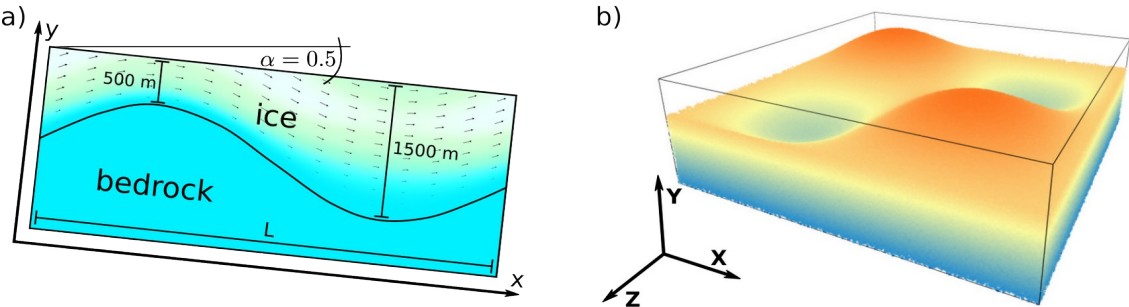

**Figure 1: (a): 2D geometry of Experiment B. This is identical to a section parallel X located at $\hat{y}$ = 0.25 in Experiment A (Right). Sloping angle $\alpha$ is given in degrees. Also depicted is the velocity field of the flowing ice, resulting for a model width $L$ of 5000 m from the simulations described below. Color and arrow length visualize the amount of velocity. (b): bedrock topography for Experiment A and general naming scheme for the axes of 3D experiments.**

### 3.3 Experiments C and D

Experiments C and D are similar to Experiments A and B, although the topography of the bedrock is flat. Instead, the coefficient of basal friction $\beta^2$ varies in a sinusoidal manner. The ice thickness $H$ is constant at 1000 m. The slope angle $\alpha$ of the ice surface and of the underlying bedrock is 0.1°. Ice flow is governed by Eq. 3 and the material parameters are summarized in Table 1. According to the benchmark specifications, Experiment C is run exclusively in 3D, and Experiment D exclusively in 2D.

The basal friction coefficient relates the basal drag $\tau^b$ to the basal velocity $v^b$ by

$$\tau^b = v^b \beta^2. \tag{8}$$

With $\omega = 2\pi/L$ the coefficient of basal friction $\beta^2$ for Experiment C is defined as:

$$\beta^2 = 1000 + 1000 \cdot \sin(\omega x) \cdot \sin(\omega y), \tag{9}$$

and for Experiment D by

$$\beta^2 = 1000 + 1000 \cdot \sin(\omega x). \tag{10}$$

Figure 2 shows the basal drag and the basal velocity calculated from Eq. (10) (Experiment D) and Eq. (8). Here, the velocity is calculated for a constant basal shear stress $\tau^b = \rho_{\text{ice}} g H \sin(\alpha)$, according to the shallow-ice approximation (SIA) (Hutter, 1983). Notice the singularity in the velocity field (Fig. 2b), which develops because $\beta^2 = 0$ at $x = \frac{3}{4} L$.

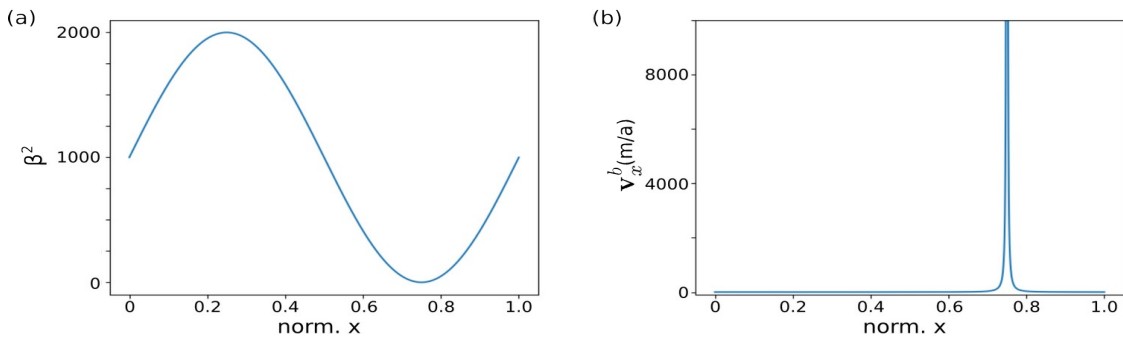

**Figure 2: (a) Basal drag $\beta^2$ and (b) basal velocity $v_x^b$ plotted according to Eq. (9) and (10) and applying the SIA, plotted for $L$ = 5000 m. Notice the singularity in the velocity field, which develops because $\beta^2 = 0$ at $x = \frac{3}{4} L$, which is within the modeled domain.**

### 3.5 Experiment E

Experiment E is a two-dimensional diagnostic experiment along the central flowline of a glacier in the European Alps (Haut Glacier d'Arolla). The basic experiment and geometry are described in Blatter et al. (1998) and by Pattyn et al. (2002). The

general geometry of the glacier profile as used in the experiment is shown in Fig. 3. The experiment is run with two different basal conditions: 1) the ice is everywhere frozen to the ground ($\beta^2 = \infty$), or 2) a zone of zero traction ($\beta^2 = 0$) between $x = 2200$m and $x = 2500$m exists. Compare Eq. (8) for the meaning of the basal friction coefficient $\beta^2$.

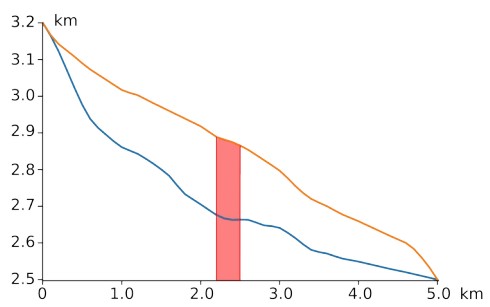


Figure 3: Longitudinal profile of Haut Glacier d'Arolla (Pattyn, 2002). Blue line: contact ice-rock. Orange line: contact ice-air. Red zone: area of varying basal conditions, with either $\beta^2 = \infty$ or $\beta^2 = 0$.

### 3.6 Experiment F

Experiment F is a prognostic experiment in which a free ice surface relaxes until a steady state is reached for a zero surface
mass balance. The slab of ice is resting on a bed with a mean slope $\alpha = 3°$. The bedrock plane parallels the surface, but is perturbed by a Gaussian bump. The initial bedrock ($B^0$) and surface ($S^0$) elevation are described by:

$$S^0(x, y) = 0 \tag{11}$$

$$B^0(x, y) = -H^0 + a_0 \cdot \left( \exp\left[ \frac{-x^2 + y^2}{\sigma^2} \right] \right), \tag{12}$$

where $\sigma = 10\,H^0$.

Experiment 'F' applies a Newtonian ($n = 1$) flow law, given in Table 1, so that the effective viscosity $\eta_{eff} = (2\,A_{n=1})^{-1}$. The experiment is run with two different values for the slip ratio $c$, so that $c = 1$ and $c = 0$ is applied. $c$ is used to describe the
basal friction coefficient $\beta^2$ (see Eq. (8)) by

$$\beta^2 = \left( c\,A_{n=1}\,H^0 \right)^{-1}. \tag{13}$$

### 4 The FE mesh

The model domain is discretized by quadrilateral Q1/P0 elements, where velocity is continuously linear and pressure is discontinuously constant. The pressure grid is offset from the velocity grid. A direct comparison of the pressure at fixed locations to the results of other models is therefore prone to some interpolation error. Periodicity of the in- and outflow
boundaries is applied in all experiments except experiment 'E'.

The ice-rock interface is defined by either of the 2 following methods, depending on the type of experiment: i) by particles, or ii) directly by the mesh geometry.

### 4.1 FE grids with a rectangular hull

In most experiments we assume that the bedrock is identical to the basal grid boundary. In experiments with a flat bedrock
topography ('C', 'D' and 'F') this means that the resulting shape of the mesh is rectangular.

An exception is the 2D-version of Experiment 'B': in order to evaluate the impact of the mesh geometry and of a particle representation of the bedrock material, we define the sinusoidal bedrock topography using particles on a FE grid with a rectangular hull. Different materials are represented by particles with different rheological properties. Particles are assigned to either ice or bedrock depending whether their depth exceeds the local ice thickness $H$. As pointed out, the material point

method can lead to spurious fluctuations in gradients at material boundaries, if particles of different materials are both located in one element.

In order to test and improve this behavior we define three different internal grid geometries and compare the smoothness of the resulting basal shear stress $\tau_b$. These geometries are (i) a classic rectangular grid, (ii) a grid, where the grid resolution increases in the vicinity of the ice-bedrock interface, and (iii) a grid, where the mesh perfectly fits the rock surface (Fig. 4).

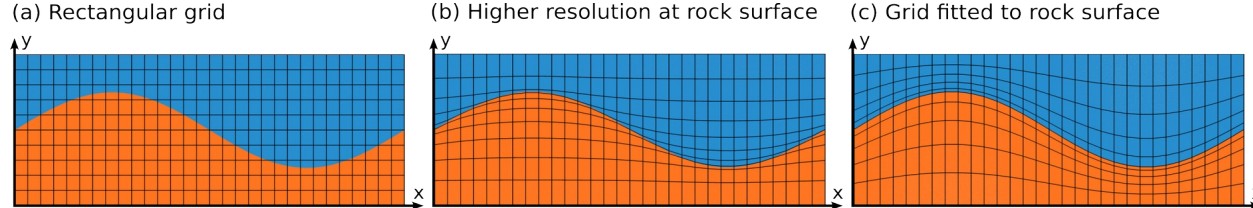

(a) Rectangular grid      (b) Higher resolution at rock surface      (c) Grid fitted to rock surface


**Figure 4: (a) Rectangular mesh. (b) Structurally conforming mesh, with increased resolution at the ice-rock interface. (c) Mesh perfectly fits the rock surface. Blue: ice, red: bedrock. For visualisation purposes, both mesh resolution and distortion are reduced compared to the actual experiments. The structure parameter $m = 0.5$ for both adapted grids (see Eq. (10), Eq. (11)). Mesh resolution: x=128, y=64. In case of the rectangular mesh additionally 256 x 128.**

### 4.1.1 Rectangular grid geometry


The benchmark assumes a constant height of the model for each experiment, while the width is varied in a series of runs during such an experiment. It is typically expected that the accuracy of the solution and the computation time are most optimal if the aspect ratio of cells is close to one.

### 4.1.2 Structurally conforming grid

We adapt the rectangular mesh to the underlying topography defined by the ice-rock boundary by vertically shifting its vertices, so that the model resolution is significantly increased close to the bedrock surface. Using the vertices of the regular rectangular grid as input: the new vertical y-coordinate $p'_2$ of a vertex point $p = [p_1, p_2]$ from the regular mesh becomes

$$p_2 = \begin{cases} s(p_1) - \Delta y \cdot \left( \dfrac{|\Delta y|}{s(p_1) - Y_o} \right)^m & if\, s(p_1) > p_2 \\[2em] s(p_1) - \Delta y \cdot \left( \dfrac{|\Delta y|}{Y_1 - s(p_1)} \right)^m & if\, s(p_1) \le p_2 \end{cases} \tag{8}$$

We assume here that the grid in *y*-direction originates at $Y_0 = 0$ and ends in $Y_1 = 1$. The rock surface is defined by a function $s(x)$. $\Delta y = s(p_1) - p_2$ is the vertical distance between the rock surface and $p_2$. Both, resolution as well as the geometry of
individual cells, are adapted to the interface line as shown in Fig. 3b. *m* is a structural adaption parameter, controlling the intensity of the mesh deformation. It is worth pointing out that the resolution in *x* is not affected by this geometry.

### 4.1.3 Grid fitted to ice-rock interface

The mesh defined by Eq. (10) does not guarantee that the ice-rock interface is aligned perfectly with finite element edges. This may still introduce stress perturbations in elements containing different materials with strong viscosity contrasts (Yang
et al., 2020). Therefore it makes sense to apply another mesh structure whose mesh edges fit the ice-rock interface exactly.

We define the new vertical y-coordinate $p'_2$ of a vertex point $p = [p_1, p_2]$ from the regular mesh by

$$p'_2 = \begin{cases} s(p_1) - (s(p_1) - Y_0) \cdot \dfrac{n_0 - n}{n_0} \left( \dfrac{n_0 - n}{n_0} \right)^m & if\, s(p1) > p2 \\[2em] s(p_1) - (Y_1 - s(p_1)) \cdot \dfrac{n_0 - n}{n_t - n_0} \left( \dfrac{n_0 - n}{n_t - n_0} \right)^m & if\, s(p1) \le p2 \end{cases} . \tag{9}$$

The variable *n* denotes the *n*th node in the vertical y-direction, and $n_0$ is a predefined node, which is relocated exactly to the rock surface. $n_t$ is the total number of vertical vertexes. The difference between Eq. (11) and (10) is that we fix $n_0$ in Eq. (11)

while $n_0$ varies along x-direction for the case discussed in Eq. (10). As before, $m$ is an adaption parameter which controls the intensity of the mesh deformation. The adaption of the grid geometry does not affect the position of nodes in x-direction.

## 4.2 FE grids with a non-rectangular hull

In all other experiments with an uneven bedrock topography (experiments 'A' and 'E' and the 3D-version of experiment 'B') we apply Eq. (11) with $m = 0$ to the lower system boundary. In case of experiment 'D', the interface ice-air is not flat, therefore particles represent ice and overlying air (Fig. 5).

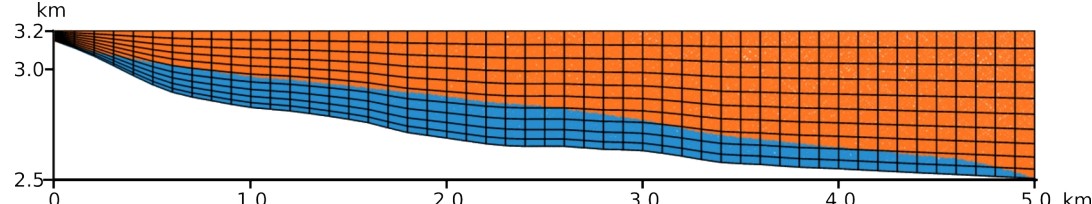

**Figure 5: Grid geometry and particle distribution used for experiment 'E', representing the glacier 'Haut Glacier d'Arolla'. Particles carry the rheological properties of ice and air. Blue: ice, red: air.**

## 4.3 Basal conditions

'Underworld' pre-implements no-slip and free-slip boundary conditions for flat system boundaries. However, experiments 'C' through 'E' require the usage of a friction law. We realize the basal drag requirement by a basal layer with Newtonian viscosity, whose viscosity is dependent on the basal friction coefficient $\beta^2$.

Following the relation between the Newtonian viscosity of this basal layer ($\eta_b$) and $\beta^2$ is derived. For the sake of simplicity, we assume a flat horizontal surface, so that the usual notation can be used. In case of an uneven base $x$ and $y$ correspond to directions parallel or perpendicular to the lower system boundary.

Combining and integrating the relations $\tau = 2\eta_b\dot{\epsilon}$ (Newtonian flow) and $\dot{\epsilon}_x = \partial v_x/\partial y$ (definition of the strain rate) leads to $\eta_b(x) = h \cdot \beta^2(x)$, with $h$ the layer height. This relation is then used to define the local viscosity of the Newtonian layer. At the top of this layer, the velocity condition defined in Eq. (8) is satisfied.

## 5 General performance tests

Below, we will first examine the CPU time consumption as a function of the grid resolution and the effect of the grid geometry on the smoothness of results.

### 5.1 CPU time consumption

The mesh resolution is one of the most important parameters that controls the precision of the solution. Since computation time increases with resolution, it is desirable to establish a relationship between these two quantities. Below we test and display the CPU time of the initial solution of the 2D-version of experiment 'B'. The computation time is not directly linked to the mesh resolution, but instead to the degrees of freedom (DOF) of the system. For 2D-Stokes problems with quadrilateral elements the degrees of freedom are $3N$ with $N$ the total number of nodes (Gagliardini and Zwinger, 2008). Figure 6 shows the relationship between the DOF and the computation time in a log-log plot for a series of 13 simulations with different resolutions and for different solvers (MUMPS, LU and multi-grid). The ratio of width to height of the grid cells remains constant in all experiments. On the hardware side, the simulations ran on a Symmetrical Multiprocessor (SMP) system with an Intel Xeon E5 processor and 32 GB RAM.

The interpolation between N and the computation time ($s$) expressed by a power regression is 0.00014 $N^{1.21}$ (multi-grid), 0.00037 $N^{1.06}$ (MUMPS) and 0.00013 $N^{1.15}$ (LU). In case of the multi-grid method outliers from the generally linear relation exist, which are related to the recursive refinement of the grid. Outliers do not exist for the direct solvers MUMPS and LU.

It is important to note, that - theoretically - the direct solvers should scale not better than $N^2$. Therefore, the power regression cannot be used in order to extrapolate these results to an arbitrary DOF. It can be seen in Fig. 6 that a slight upward curvature of the data with regard to the interpolation exists.

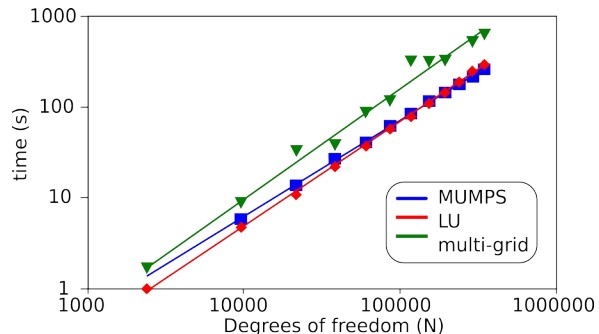

**Figure 6: Computation time for Experiment B plotted vs the degrees of freedom and using different solvers. Blue: Mumps (0.00037N$^{1.06}$), red: LU (0.00013N$^{1.15}$ ), green: multi-grid (0.00014N$^{1.212}$).**

## 5.2 Impact of the grid geometry

We tested the impact of the grid geometry based using the flow law parameters $n$=3 and $A$=10$^{-16}$ Pa$^{-3}$a$^{-1}$ (Pattyn et al., 2008). In order to increase effects related to the geometry, the mesh resolution is intentionally relatively small with 128x64. Only for the rectangular grid, a larger resolution is applied. The non-rectangular meshes use an adaption parameter m=0.25.

The results for the surface velocity are generally smooth and virtually identical in the three grid geometries. However, fluctuations of the shear stress can become considerable, especially close to the ice-bedrock interface ( Fig. 7), depending on the type of FE grid. These fluctuations are largest if a low-resolution rectangular mesh is used. Increasing the resolution of the rectangular mesh does not fully eliminate the fluctuations, but is a way to reduce the problem ( as highlighted in Fig. 7c). A more conforming grid with an increased resolution around the rock surface (Eq. 8), reduces the fluctuations significantly, but does not completely eliminate them (Fig. 7b). The unambiguously smoothest results are achieved using a grid fully fitted to the rock surface after Eq. (9 (see Fig. 7a). This confirms that fluctuations become smaller when there is less mixing of materials with different properties within an element.

The absence of fluctuations in the stress field is a measure for the ability of the model to deal with discontinuous material boundaries. In the following simulations for Experiment B, we will therefore exclusively apply the fitted mesh. This is in line with the original benchmark paper by Pattyn et al. (2008), who suggested to test the models with optimized settings.

The boundary between ice and the bedrock is usually the only non-planar material boundary in glaciers and ice sheets. The other material boundary, between ice and air, can often be considered almost planar in the modeling of large ice sheets, while rheological changes within the ice itself can usually be treated as gradual, controlled for instance by the temperature field or the crystallographic fabric. It is an important conclusion from these results that a FE mesh fitted (even approximately) to the underlying bedrock topography can significantly improve the accuracy of ice flow simulations based on the material point method.

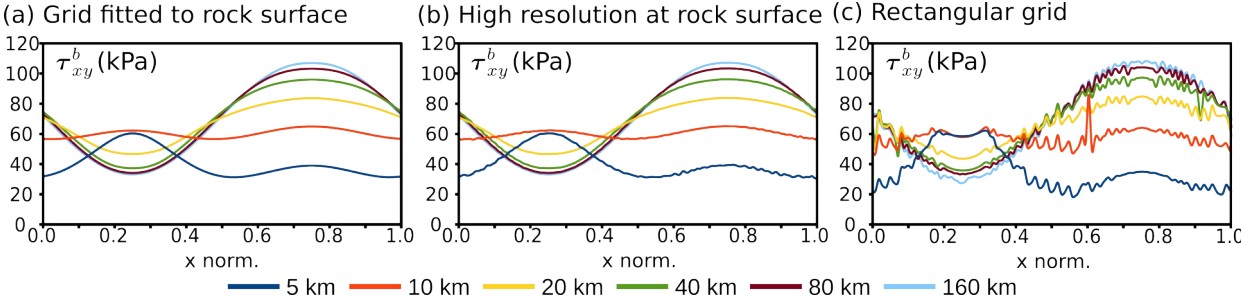

**Figure 7: The basal shear stress $\tau_{xy}^{b}$ in kPa, calculated on (a) a grid fitted to the rock surface, (b) a structurally more conforming grid, which increases mesh resolution at the rock surface and (c) a rectangular grid. The experiments are based on a grid resolution of 128 x 64.**

## 5.3 Impact of the grid resolution

Choosing a well suited mesh resolution is always a balance between precision requirements and computational limitations. The computation limits can be related either to the hardware or to general time constraints, which do not allow very long computation times. Also mesh geometry has an impact on the accuracy and may or may not require a larger resolution.

We performed the 2D experiments on a classical SMP workstation (Sec. 5.1). Given the existing time constraints and the number of simulations this allowed us to test resolutions of 64 x 32, 128 x 64 up to 256 x 128. Even at the relatively small resolution of 64 x 32 the solution for stress and velocity at the surface of the model are smooth and virtually identical to the results obtained with higher resolutions, independent of the mesh geometry (Sec. 5.2). This was different in case of of the interface between the ice and the underlying bedrock. While performance was well enough or very good with a resolution of

128 x 64 for grids adapted to the interface, a resolution of 256 x 128 was necessary to obtain usable results in case of a rectangular mesh. Two grid points above the interface the solution was smooth for all mesh geometries at 128 x 64 and didn't differ from higher resolutions.

Highest resolution 3D simulations ran mostly on a high performance computer cluster. Accuracy was systematically tested for Experiment B using a fitted grid, by comparing the results of the 2D and 3D experiments. Under perfect conditions 2D

and 3D simulations which should yield identical results. We found that the necessary resolution is highly dependent on the system size, i. e. on the aspect ratio of the cells. For small system sizes of up to 20 km we found that a minimum resolution of 256 x 128 x 32 was necessary to produce similar results as the 2D model. Larger systems produced satisfactorily results already with a resolution of 128 x 64 x 8.

## 6 Specific results

Below we will compare the results of the individual benchmark experiments generated by Underworld to the results of alternative codes. Where applicable we run both, the three-dimensional and the two-dimensional version of the benchmark experiments. This applies to Experiments 'B' and 'C'. For the latter, Experiment 'D' is the associated 2D-setup. All results are compiled in the supplementary material to this paper. A note on the output-parameter $\Delta p$, which is the difference between the isotropic and the hydrostatic pressure: the curve progression is very similar to that of other Full Stokes model of

the ISMIP-project, but show stronger fluctuations between adjacent evaluation points. This is due to the internal architecture of Underworld-experiments, where the mesh used for the calculation of pressure is a sub-mesh of the velocity-mesh with a staggered geometry. Fluctuations are a product of internal interpolation.

### 6.1 Experiments 'A' and 'B'

We fit the lower system boundary to the topography of the underlying bedrock in the 3D experiments by applying Eq. (11)

with $m = 0.2$ to the lower system boundary. The 2D version of experiment 'B' is based on a rectangular grid, which is internally fitted to the rock surface as in Sec. 5.2 above. Other relevant parameters are given in Table 1.

Results of experiment 'A' are shown in Fig. S1 and S2 of the supplementary material, which display the surface velocity and the basal shear stress. Results of Experiment 'B' are shown in Fig. S3 to S6 of the supplementary material, showing surface velocity and basal shear stress in a 2D version of the experiment and at an arbitrary section paralleling the X-axis of the 3D

experiment. Diagrams include the results of Full Stokes solutions published in Pattyn et al. (2008) for comparison.

The surface velocity is controlled by the model width $L$ and is in the range from a few $m$ a$^{-1}$ to more than 100 $m$ a$^{-1}$. Figure 8 compares our results for Experiment 'B' to results compiled from 8 full-Stokes models that were previously published (Pattyn et al., 2008). A full comparison of the results is in the supplement to this paper.

In Experiment 'B', the shape of the horizontal surface velocity for $L = 5$ $km$ differs significantly from that for the other

cases. The surface velocity is larger over the bump and thus anti-correlated with the ice thickness. Gagliardini et al. (2008) explain this as a mass conservation effect: horizontal flux cannot be balanced by vertical flux, because the vertical velocity would be to large for the given system size. This phenomenon does not occur in Experiment 'A', with the same system size, although the section is identical at the chosen location. This can be explained because ice can flow around the sides of the bump.

Figure 8 shows the maximum and minimum surface velocity $v_x(y_s)$ and the maximum basal shear stress $\tau_{xy}(y_b)$ calculated by the 2D version of Experiment 'B', and compares it to other Full Stokes solutions compiled by Pattyn et al. (2008). Maxima of both the shear stress and of the surface velocity show a tendency to be at the lower end of the spectrum, while minima are virtually identical to the results from the comparison models. However, a full comparison (compare supplement) shows a more diverse distribution of the results than the minima / maxima comparison in Fig. 6 implies. The results of our

Experiments show generally a very good agreement with the majority of models, while some of the comparison models display large deviations with regard to the entire curve.

Figures S5 and S6 of the supplementary material compare the results of the 2D and the 3D setup, which should be ideally identical. The results for the surface velocity are virtually identical, with exception of the $L = 5\,km$ case, where a deviation of ~ 2 % for the maximum and the minimum velocity exists if comparing 2D and 3D results. A similar effect exists regarding the extrema of the basal shear stress, which are most notable for $L$=5 km and $L$=10 km.

To shed some light on the difference between 2D and 3D results, we ran 3D simulations with a resolution of 128 x 64 x 8 and of 256 x 128 x 32 for these system sizes and compared them to the results of the 2D simulation (256 x 128). Regarding the surface velocity, the higher-resolution 3D mesh yields indeed results closer to the 2D result than the lower-resolution simulation (Figure S6 of the supplementary material). This is not the case for the basal shear stress, where the deviation from the 2D result is comparable for both resolutions.

Possible reasons are: (a) different hardware, the 2D experiment as well as the low-resolution 3D experiment ran on the same machine, while the high-resolution 3D experiment ran on an HPC-cluster. And (b) differences in the mesh geometry. The low-resolution simulations use an exponent $m = 0.25$ in Eq. 9. For technical reasons $m$ is set to 0 for experiments on the cluster.

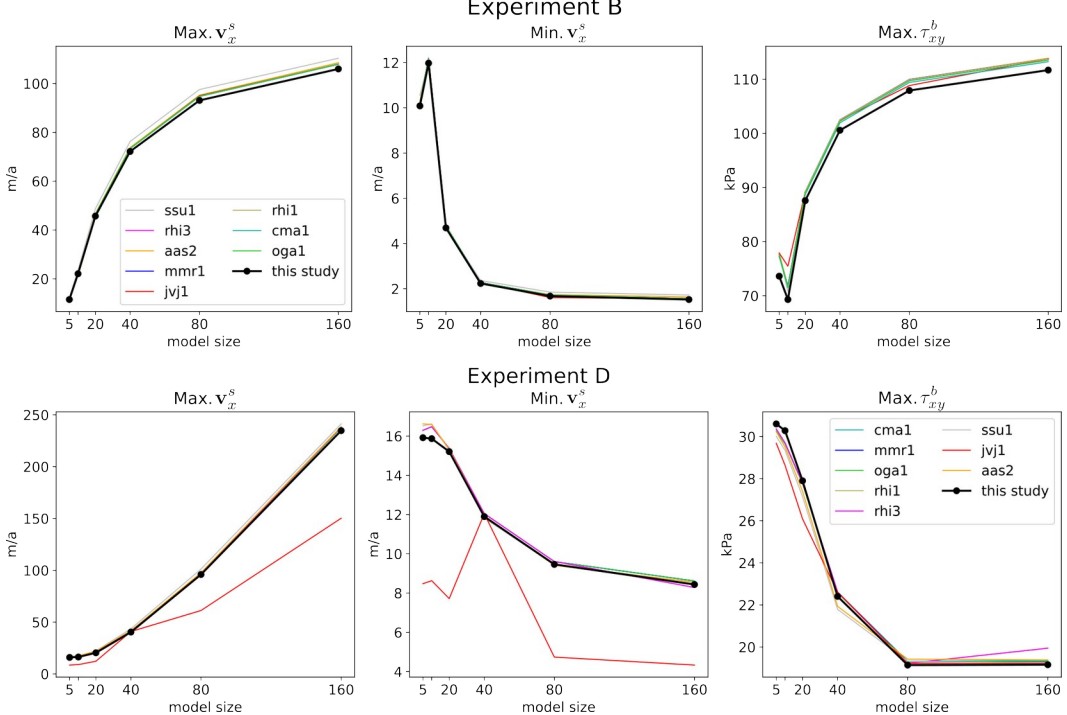

Figure 8: Maximum and minimum values of the horizontal surface velocity ($v_x\left(y_s\right)$) and of the maximum basal shear stress ($\tau_{xy}\left(y_b\right)$) in Experiment B and D, plotted for every model size. The underlying FE mesh is fitted to the rock-ice interface. Results are compared to the results of 8 Full Stokes models (compiled and published by Pattyn et al., 2008). Abbreviations of the comparison experiments are compiled in Table 3.

| Model | Dimensions | Method | Reference |
|---|---|---|---|
| tsa1 | 2 | MPM | this study, based on Mansour *et al.* (2020) |
| Jla1 | 3 | MPM | this study, based on Mansour *et al.* (2020) |
| aas2 | 3 | FE | unpublished |
| cma1 | 3 | FE | Martín et al. (2004) |
| jvj1 | 3 | FE | Johnson and Staiger (2007) |
| mmr1 | 3 | FE | unpublished |
| oga1 | 3 | FE | Gagliardini and Zwinger (2008) |
| rhi1 | 3 | Sp | Hindmarsh (2004) |
| rhi3 | 3 | Sp | Hindmarsh (2004) |
| ssu1 | 2 | FE | Sugiyama et al. (2003) |

Table 3: Model abbreviations used in the diagrams, taken from Pattyn et al., 2008. 'Dimensions': model dimensions. 'Method': numerical method: FE = finite elements, Sp = spectral method, MPM = material point method.

### 6.2 Experiments 'C' and 'D'

Parameters for Experiments 'C' and 'D' are given in Table 1 for *n=3*. Results are summarized in Fig. S6 and S7 (Experiment 'C') and Fig. S8 and S9 (Experiment 'D') in the supplementary material.

In Experiment 'C' the surface velocity is strongly dependent on the system size *L*. In case of the smallest system size (*L*=5 *km*) the surface velocity is close to constant at ~16 *m* a$^{-1}$. With *L*=160 *km* it ranges from 8.8 *m* a$^{-1}$ up to 122.5 *m* a$^{-1}$ (Fig. S7, supplement). The shear stress results of the Underworld-simulations line up well with the comparison simulations. They show a sinusoidal curve with a maximum at $\hat{x}$=0.25 and a minimum at $\hat{x}$=0.75. With increasing model size, the maximum gets progressively flattened. The peak downwards at the singularity value $\hat{x}$=0.75 is far less impacted by model size, which means it gets more dominant for bigger values of L.

In Experiment 'D', the maximum surface velocity increases with model width *L* and is in the range from 16 *m* a$^{-1}$ to more than 235 *m* a$^{-1}$. Figure 9 shows the maximum and minimum surface velocity and the maximum shear stress $\tau_{xy}$. Results are generally in good agreement with the majority of model results they are compared to. However, the general variation between the results of the comparison models is generally larger than in the 2D version of Experiment 'B', the only other 2D experiment of the benchmark. The same statement applies to basal shear stress. Again, a full comparison of the results is included in the supplement to this paper.

### 6.3 Experiments 'E' and 'F'

Experiment 'E' is calculated for two different setups, where either the entire glacier is frozen to the underlying bedrock, or where a traction-free section close to the center exists. In both cases, the results of Underworld lies within the range of the comparison models (see Fig. S10 and S11 in the supplementary material). Both the calculated surface velocity and the basal shear stress are in the lower range of this range, which is true for the majority of models. In particular extreme values of the curve are a bit smoother than in some of the other Full Stokes models.

The prognostic Experiment 'F' calculates the surface velocity and the surface topography. Results are compiled in Fig. S12 and S13 in the supplementary material. Since the majority of potential comparison models are not capable to calculate a free surface, the results of Underworld can be compared to only two other software packages. Due to the special properties of the software, we model only the version of the experiment with a basal no-slip condition. Other values for the basal traction would either involve a very complex implementation or yield questionable results. We ran the model until no further change in topography or velocity occurred, and interpreted this as the stable state.

When we compare our results with those of the two reference models with respect to an analytical sample solution (Frank Pattyn, 2022, personal communication), qualitatively similar results are obtained. One of the reference models shows very good agreement with the theoretical surface elevation, but shows lower accuracy with respect to surface velocity. Underworld predicts the surface velocity the best of the three models, but tends to develop a slightly less extreme topography than the analytical solution.

### 7 Comparison experiments with anisotropic and isotropic ice

Ice is assumed to deform mostly by dislocation creep at natural strain rates (Budd and Jacka, 1989), whereby slip along the basal planes is much easier than slip along the other slip planes (Duval et al., 1983). This makes an ice single crystal mechanically effectively transversely isotropic, with the c-axis that is oriented perpendicular to the basal plane as the symmetry axis. The anisotropy of ice single crystals leads to a crystallographic preferred orientation (CPO) in a deforming aggregate of ice crystals (Alley, 1988; Llorens et al., 2017). The mechanical anisotropy of an aggregate may be of a lower symmetry than that of a single crystal but can be approximated to a first order as orthotropic (Gillet-Chaulet et al., 2006).

One of the features of Underworld, which makes it interesting for the modeling of ice, is its capability to model linear orthotropic viscosity, which includes transverse isotropy as a special case.. In order to demonstrate its effect we set up comparison experiments with anisotropic and isotropic ice rheology, based on the setup of Experiment 'B': ice flows over a sinusoidal surface, driven by a general 0.5° tilt of the model. Isotropic flow is governed by Eq. 4, using the parameters given in Table 1.

The orientation of the anisotropy is stored as the c-axis orientation on the level of the particles in Underworld. The aggregate anisotropy of one mesh element is calculated from the individual c-axis orientations of the cloud of particles in an element. Fabric evolution is simulated using the rotation of the c-axes in the flow field after e. g. Gillet-Chaulet et al. (2005) or Richards et. al. (2021). However, for simplicity, in our example here we assume that all basal planes are and remain

horizontal. The anisotropy of ice, in terms of the ratio of resistance to slip parallel to crystallographic non-basal and basal planes, is about 60-80 in a single ice crystal (Duval et al., 1983). However, as we here simulate aggregates of crystals, we set the viscosity for shear non-parallel to the basal planes only ten times higher than the vicosity for shear parallel to the basal planes, which we assume equal to that used for the isotropic flow law.

Figure 9 shows the shape of marker lines prior to deformation and after 750 a of flow for both, iso- and anisotropic ice
models. Marker lines inherit their sinusoidal shape from the shape of the underlying topography and are then folded according to the localization of the highest shear rates. In isotropic ice (Fig. 9 b) the axial plane of the shear fold is mostly controlled by the underlying topography. In case of anisotropic ice (Fig. 9 c), the folding is more intense and the axial plane is sub-horizontal, showing that the anisotropy has a distinct effect on resulting fold structures.

Figures 10 and 11 that show the velocity field and the strain rate field in both materials allow a better understanding of the
deformation process. In case of isotropic ice, the flow field is controlled by the underlying topography. Hence the hill on the left acts as a bottleneck for the ice flow, and the hill sides funnel ice in and out of the bottleneck region. Consequently, the zone of high shear strain at the ice-rock interface extends lang the crest of the bedrock bump (Fig. 10 a) and the maximum velocity above it. Outside the bottle-neck region flow is comparatively evenly distributed.

In the case of anisotropic ice, the flow regime and thus the shear folding is quite different. Here, flow is strongly controlled
by the inherent anisotropy and far less by the bedrock topography. Looking at the velocity field (Figure 11 a), it becomes clear that the flow field is internally subdivided into a fast flowing upper part and slow flowing 'dead ice' in the lower part. Decoupling of the shallow and deep ice develops because the anisotropy facilitates shear along the horizontal basal planes. The resulting horizontal high strain zone spans the entire system and produces the distinct shear fold.

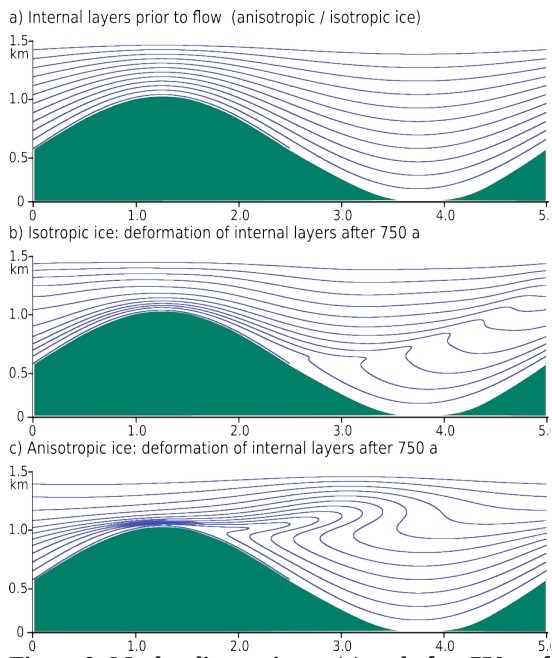

**Figure 9: Marker lines prior to (a) and after 750 a of flow of (b) isotropic and (c) anisotropic ice. The axial plane of the resulting shear fold in isotropic ice mimics the bedrock topography, while it is controlled by shearing along a horizontal shear zone in case of anisotropic ice. Green: bedrock, flow to the right.**


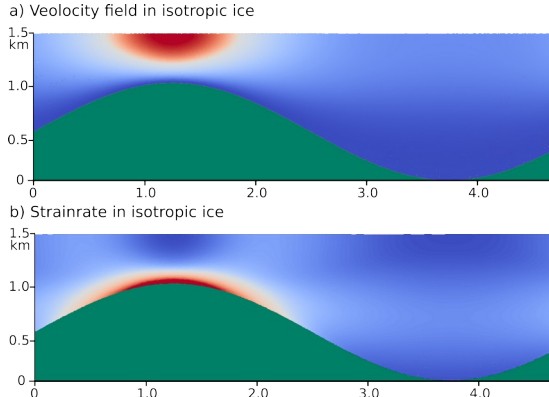

**Figure 10: Velocity field and strain rate field in isotropic ice. Large strain rates and velocities occur in the vicinity of the bottleneck formed by the crest of the hill. Green: bedrock, velocity: red = 70 m/a, blue = 0 m/a, strain rate: red = 0.032 / a, blue = 0 / a.**

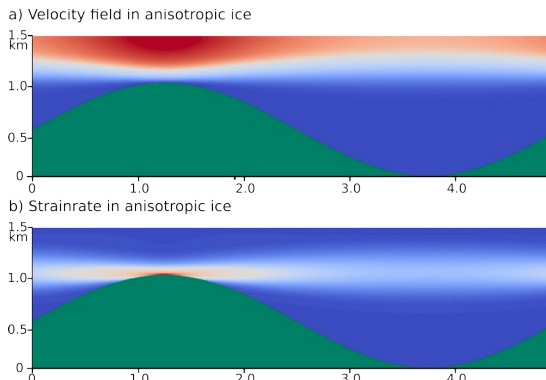

**Figure 11: Velocity field and strain rate field in anisotropic ice. The velocity field is vertically subdivided into a fast flowing upper and an almost stagnant lower part. The strain rate is thus at its maximum in the horizontal shear zone that spans from crest to crest of the bedrock undulations. Green: bedrock, velocity: red = 6.6 m/a, blue = 0 m/a, strain rate: red = 0.031 / a, blue = 0 / a.**

## 8 Outlook

One of the great advantages of the Underword2 software package from the standpoint of the modeler are its great flexibility and traceability due to its hybrid nature as both particle and Finite Element model. Another contribution to its flexibility is the easy extensibility of the core code and the interactive development thanks to the existence of a Python API. The compatibility of the API with the mathematical-scientific packages NumPy and SciPy provides easy access to a wide range of numerical resources. Anisotropic flow and heat flow are already part of the core package.

Given the current debates in the ice community about the role of the anisotropy and hence of the fabric evolution of ice, or the ideas for improvements of the flow laws for ice (e.g. Kennedy et al, 2013; Llorens et al., 2017; Richards et al., 2021; Martín et al., 2009), this type of flexibility is an important precondition for a future proof numerical model for glacier ice. One example was the implementation of the orthotropic rheology and the fabric evolution in deformed ice following Gödert (2003) and Gillet-Chaulet et al. (2005) by the authors of this text. Possible improvements to the fabric evolution models - for
instance by the SpecCAF model (Richards et. al, 2021) - can be easily implemented and tested.

A few desirable features are currently still missing but are on the road map for future versions of Underworld. The soon-to-be released Underworld3 for instance allows greater flexibility of the mesh geometry, including the triangulation of shapes and areas with arbitrary geometry.

**6 Conclusions**

The Underworld software is designed to solve deformation in complex geodynamic systems with non-linear elasto-visco-plastic materials, for which it provides a full-Stokes solution. It is therefore well suited for the modeling of glacier and ice-sheet flow, as it includes heat flow and anisotropic rheology. The combination of Lagrangian mass points (particles) and a Eulerian Finite Element solution allows the tracking of individual points as well as of inner and outer surfaces, such as deforming stratigraphic layers, but also of the thermal-mechanical properties in deforming materials. In case of large rheological differences across interfaces, the possibility to fit the grid to the interface greatly improves the accuracy of stress field, compared to other grid types. In case of ice flow experiments, it makes sense to fit the grid to the bedrock-ice boundary.

We compared results of Underworld simulations with those of other modeling approaches for the set of benchmark experiments provided by Pattyn et al. (2008). Our results match the full-Stokes solutions that are compiled in that study. This means that Underworld is a viable alternative to other full-Stokes models, in particular where the Material Point Method is advantageous, such as when accurate tracking of material volumes or stratigraphic layers is desired. A further advantage is that, owing to the built-in Python API, Underworld is very flexible and can be extended to be applied to even the more complex processes which are involved in the flow of ice sheets and glaciers.

**Code/Data availability**

- The program code that defines the experiments discussed in this article is included in the supplement to this article.

- Graphical representations of the results as well as data files are found in the supplementary material to this article.

- In addition to the supplement, the complete data set is also available here: https://gitlab.com/underworld2/ISMIP-HOM-Data.

- Source code for Underworld is hosted here: https://github.com/underworldcode/Underworld2

**Author contributions**

T. Sachau and P. Bons discussed and designed the first implementation of the benchmark experiments. H. Yang provided the code and the concept behind the applied mesh geometries. T. Sachau and H. Yang developed and tested the code, and performed some simulations. J. Lang performed most of the 3D experiments, compiled the related data files and data plots. T. Sachau performed the 2D simulations, compiled the related data files and data plots. T. Sachau prepared the manuscript with many contributions, corrections and revisions from all authors. P. Bons and L. Moresi provided important guidance.

**Competing interests**

The authors declare that they have no conflict of interest.

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
