# Peer review of "ISMIP-HOM benchmark experiments using Underworld"

_EGUsphere, 2022_

## Author Response (AR1)

Referee #1, Frank Pattyn

The paper describes a particular application of the Underworld model. Underworld2 is an open-source, particle-in-cell finite element code tuned for large-scale geodynamics simulations, which allows for the tracking of history information through the high-strain deformation associated with fluid flow. In this paper, Underworld is applied to large-scale ice flow and compared to known solutions given by other full-Stokes model following the benchmark experiments given in Pattyn et al. (2008). The authors demonstrate that the results of their model is in overall agreement with results for other full-Stokes models, which shouldn't be surprising. The major differences are related to different numerical approaches that are imbedded in Underworld and the authors show what numerical approach is more appropriate for this particular application.

While this is in itself an interesting model application, to me this is a stepping stone towards a more in-depth study of ice dynamics/mechanics and especially ice rheology. Indeed, Underworld is a far more sophisticated model than the application to an isotropic ice mass without taking into account kinematic constraints (the geometry is fixed for most experiments with exception for experiment F). The authors mention a series of potential applications in the introduction, but it remains a limited list. In the outlook section, the potential is again given for improving flow laws of ice, but it remains limited to this (radar stratigraphy isn't mentioned anymore). A more exhaustive discussion of where potentially Underworld can be applied and how it can improve our understanding of ice mechanics, rheology, fabric, etc. would be more than welcome. Especially the link with ice core and seismic/radar studies are of importance. For instance, studies based on phase-sensitive radar (pRES) allow for quantifying COF in ice sheets using polarimetry and could greatly profit from the full capability of Underworld (e.g., Ershadi et al. (2022) and reference herein; Drews et al., 2015). Other examples are the elastic and visco-elastic response on short time scales due to drainage of supraglacial lakes within ice shelves, which have an effect on the ice shelf rheology (Banwell et al., 2019; MacAyeal et al., 2021), to name a few. Strengthening and emphasising this part is important, as Underworld has certain advantages over other ice sheet models, but also certain limitations. Ice sheet models, and definitely the majority that participated in ISMIP-HOM, are designed to make prognostic runs of ice sheets and glaciers across different time scales. Certain approximations are therefore made to make them computationally efficient and their focus is often on more complex boundary conditions to deal with atmosphere and ocean interaction, for instance. Underworld can be used to study certain aspects of ice dynamics, as the examples cited above. This should be stressed and strengthened.

We thank the reviewer for the overall supportive and helpful comments and tried to address them all. Modelling of flow of ice has many applications, of which we only mentioned a few, admittedly biased by our own research area and intended application of Underworld. As suggested by the reviewer we have expanded the references and mention the radar work, including new possibilities to detect anisotropy more prominently. The main intention of this paper is to show that Underworld is on par with other full Stokes codes. The rationale behind is that in our experience this is what the community expects to be shown. We therefore originally refrained from presenting new simulations. However, in light of both reviews we now added a whole section 7 in which we illustrate that Underworld can relatively easily incorporate mechanical anisotropy and that it has a significant effect on the distribution of flow within an ice volume. We refrain from other examples, but the reader is referred to examples of various published applications to geodynamics. We hope that readers will be consider Underworld when seeking the best code for their specific problem, although this will certainly not always be Underworld.

Minor comments:
Line 48: The correct website: https://frank.pattyn.web.ulb.be/ismip/welcome.html (12/7/2022), but all material is also found on the TC website, which is maybe better to cite.
The link is changed and reference is made to the supplement of the TC-paper

Figure 4: I guess these are examples of the type of mesh and not the actual meshing used (it seems rather coarse to me). Mesh resolution is discussed later, but it would be informative to express the different mesh sizes used (not only in number of degrees of

freedom, but also in number of x,z or x,y,z).

We included the mesh size in the caption. We hope this makes it clear.

Line 249: This is the only mention of x,z mesh size in the paper. What are other used? Do you think the highest resolution employed is sufficient to solve the problem? This can be tested by comparing the results for different resolutions and see whether or not this converges. What is the highest mesh resolution used? Is this sufficient for the purpose of this study and how may this potentially hamper other detailed studies of ice dynamics?

Thanks, that is an important question. We included a sub-section about mesh resolution as section 5.3. We hope this answers the questions satisfactorily. Mesh resolution is a complex matter, as models in 2D and 3D with different system sizes and several internal geometries exist. At least, we hope that this can give some orientation for other modelers about what they can expect.

Line 345: You should essentially compare your result with the analytical solution rather than the numerical solution of two other models. Having said that, I realised that the analytical solution is not given in the repository of the ISMIP-HOM results. I added the matlab file from my archives that can be used for this purpose.

We included the analytical solution in the comparison. We cited this as 'personal communication' with you.

Conclusions: Please refrain from a bulleted list and write a section in plain text.

We removed the bullet points and reordered the conclusions into two paragraphs.

References

Banwell, A.F., Willis, I.C., Macdonald, G.J. et al. Direct measurements of ice-shelf flexure caused by surface meltwater ponding and drainage. Nat Commun 10, 730 (2019). https://doi.org/10.1038/s41467-019-08522-5

Drews, R., Matsuoka, K., Martín, C., Callens, D., Bergeot, N., and Pattyn, F. (2015), Evolution of Derwael Ice Rise in Dronning Maud Land, Antarctica, over the last millennia. J. Geophys. Res. Earth Surf., 120, 564– 579. doi: 10.1002/2014JF003246.

We have extended the text to specifically refer to radar measurements and added a number of references we thought most appropriate.

Referee #2, Anonymous

The paper presents an application of the Underworld software, initially developed for geodynamical applications, to solve ice flow problems using the Full Stokes formulation. Underworld uses the material point method and shows interesting potential in addressing challenges related to ice flow modelling. The paper mostly reports about benchmarking Underworld using the ISMIP-HOM benchmark experiments. Besides the benchmarking exercise, the paper provides some technical details related to the treatment of sharp boundaries and solver performance. Efforts towards better resolving physical processes with the respect to ice flow are valuable. Underworld shows potential to include complex rheologies and exposes both a mechanical and thermal solver, for both 2D and 3D configurations.

We appreciate this summary.

Although reporting overall good results for the selected benchmarks, the paper is missing some results and discussion about the actual and claimed new and interesting features Underworld could actually handle. These features such as incorporating anisotropy, complex rheologies or thermo-mechanical coupling, points advanced already in the abstract, would truly provide a step forward in our understanding of complex processes affecting ice flow. To turn the current paper draft into a contribution making some impact, I would be very enthusiastic about seeing actually Underworld addressing some of the challenges (e.g. as listed on Line 71) and discussing them.
I would thus suggest, besides addressing the specific comment listed hereafter, to add a couple of examples that actually demonstrate the features a 2D and 3D MPM code can provide, with

Both reviewers addressed this point, and we here refer to the reply to reviewer #1. Here we wish to add that the capabilities of Underworld are already described in the literature. The applications so far were, however, not in glaciology, but geodynamics. Showing that the same routines and capabilities can also be applied in glaciology is possible, but would greatly divert from the main issue, which is benchmarking, as one the run into questions and debates on the appropriate parameters and processes to implement. Examples would be whether to use n=3 or 4? Or both? Or, what is the strength of mechanical anisotropy? All these issues apply to any code that is chosen to solve a question. Here we merely want to show the community that an additional tool is available to them and that it at least passes the basic benchmark tests that are available. To provide a glimpse of the possibilities that Underworld makes available, we added a section with a simulation with anisotropic ice. We impose an anisotropy and do not claim it is the correct one, not even that the situation is realistic. We merely show that it can be implemented, that it immediately dramatically changes the results, and, finally, that Underworld is particular useful to track the distortion in stratigraphy that develops over time. Although only 2D, we hope this suffices to prove the point.

We thank the reviewer for the careful detailed comments and tried to address them all.

Specific comments
l.10: "solution for elasto-visco-plastic materials and includes mechanical anisotropy."
that's exciting! Please provide some example as this is the main motivation of using Underworld.
We now added one example with three figures in a new Section 7.

l.13: "Furthermore,we" -> Furthermore, we. Done
l.19: "ice 1h" is not an abvious concept. consider providing some additional information or making it clear that you are defining "1h".
We changed the wording a little, hopefully this is clearer now. We think that the name 'ice 1h' for the ice polymorph found on Earth is known well enough and needs no defining.

l.24: The recent development on "complex rheologies" goes beyond the 3 articles cited here. Consider including recent work by Ranganathan and Minchew (see http://glaciers.mit.edu/publications).

We added the reference to Ranganathan, M., Minchew, B., Meyer, C. R. and Peč, M.: Recrystallization of ice enhances the creep and vulnerability to fracture of ice shelves. Earth and Planetary Science Letters, 576, 11721, https://doi.org/10.1016/j.epsl.2021.117219, 2021.

l.31-32: Maybe even more important, the strong dependence of ice rheology, i.e. viscosity, on temperature… Very true, but we assumed that this is well known and focussed here on the difficult issue of dealing with the strong anisotropy of ice 1h. All minerals deformaing by dislocation glide processes are, of course, anisotropic, but ice 1h is particularly so. Still, we now added a sentence at the beginning of the paragraph, to also highlight temperature dependence.

l.59 & Eq.1: If you have both σ and P, then your σ actually stands for the deviatoric stress tensor, usually denoted as τ. If you want to consider the full (and not "absolute") stress tensor, then you should remove the pressure derivative in Eq. (1). Please correct.
Done - removed the word 'absolute, which really had no place there. We want to consider the deviatoric stress here.

Eq.4: η_ice, consider using italic text only for math variables. Here ice is not a variable thus the italic should not be used. Please correct the other math notation for consistency.
Thanks, this is done.

Table 2: "basal shear stress parallel x", what does parallel x stands for? Please precise.
Done – we changed the wording on the Table to make clear, that we are interested in the horizontal shear stress at the basis of the ice sheet in x (= flow) direction.

l.71: "Some of these challenges" consider including some of these in your results (see general comment).
With the new section 7, we now added and compared an anisotropic and an isotropic version of Exp B.

l.216: Basal condition section. The solution of adding a viscous layer is not the most proper implementation of basal boundary condition. Since the ice-bedrock interface is crucial, it would very interesting to see how the proposed ad-hoc boundary condition implementation performs with respect to a more serious implementation of traction boundary conditions.
This is an interesting suggestion, but we think beyond the scope of this paper and would require appropriate benchmark experiments. We realise that the solution may seem 'ad hoc' here. The solution is well established in geodynamic modelling to describe the interface or suture between tectonic plates that move past each other, for example in subduction zones. We admit that further investigation of the pros and cons of this solution may be helpful.

l.229: CPU time consumption section. It would interesting to get slightly more information in this section, namely regarding the hardware used as the SMP system (l.237) is not the most common processor one would have. Also, it would be interesting to know how much RAM the compute server had.
Done – added some hardware information.

l.233-234: The number of DoFs depends on the element type also.
We mention now that we are talking about quadrilateral elements

l.242: You only report 2D scaling and performance. What about 3D? Any data available. This would be very interesting as well in order to compare.
For practical reasons we excluded 3D experiments from this specific series of experiments. They were to time consuming on the given hardware. We take this comment as a suggestion for a supplement to a future publication if based on 3D simulations.

l.255: "cp Fig. 7c", what's cp?
'compare', but we now write ' as highlighted in'

l.273: Section Specific results. What's the logic behind having some results in SI, while others in the article. I guess it's fine doing so, but it would be good to state about the strategy as it is not obvious as such to the reader. Also, why not having all or most figures in the main paper, potentially as appendix. This would be much more valuable.

The inclusion of most figures in the Supplement is guided by the organization of comparable publications that perform these specific benchmarks, following Pattyn et al. (2008). In fact it would be easier for readers of the main text if the referenced figures were included right there, but for readers interested mainly in the supplement it would be a bit more complicated. In the end, we tried to present the data as systematically as possible and hope that this is the least annoying way overall.

l.277: complimentary" -> "complementary" I guess.
Changed to 'supplementary'

l.282: Experiments B. Experiment B 3D shows not a so good fit in the Figure S5 at 5km,
and in Figure S6 for 5 and 10km. Please provide more information on why, or try to get
the discrepancy fixed.
We increased the resolution of the 3D simulations in Figures S5 / S6 for 5 and 10 km and achieved a partially better fit now. 3D simulations of this resolution have not been possible at the time of the previous version of the article. We were still working on a containerized version of Underworld that would run well enough on our HPC cluster back then.

However, we are still not completely happy with the results in Figure S6. While the velocity of the 3D simulation with 5 km is closer to the 2D result now, the shear stress at the basis is – albeit different from the previous low-res simulations – still not identical to the 2D result.

We discuss this in the Section about Exp B in the main text, starting line 346, where we give possible reasons (different hardwar + different mesh properties). We are aware that more systematic tests would be possible on this issue. But since 3D simulations are very time and resource intensive and we have a relatively wide variety of possible options (mesh resolution, mesh geometry, hardware, solvers and more), we limited the discussion to what we have for practical reasons.

l.327: "Fig.9shows" -> add space.
Thank you. Figure now written out in full as it is the first word of the sentence.

l.353: "Underworld2" where does Underworld2 come from. Until now, you always referred
to Underworld.
Thanks for noticing. Removed the '2', which is the number of the major version and is often used in the communication with the developers. But the official term of the software is just 'Underworld'.

l.367: Section conclusion. please refrain from using bullet point list.
Already done in response to reviewer #1

l.378: Here as well, potentially very exciting, but nothing is shown in this paper. Consider
adding material regarding these features.
Showing tracking of material points or layers was not part of the benchmarking experiments, so originally not included. However, with the added simulation (Section 7), we now provide an example of this.

Figures:
For all figures, consider:
- adding axes information
- using bigger font
- homogenising the style
- using panel names (a,b,etc) in descriptions instead of left, right
- Work on the overall style
We reviewed and improved the figures according to your suggestions. Thanks!

---

## Author Response (AR2)

Dear authors,

Based on the comments raised by the reviewers to your article and upon my own reading, I will be happy to consider publication upon the required technical corrections annotated by the reviewers. For your convenience I summarized the reviewer's points below.

1. Eq.(1) is inconsistent with description and Eq.(3); σ (sigma) should be replaced by τ (tau) in Eq.(1) as both stand for the deviatoric stress tensor. If σ would stand for the total stress tensor, then it would include pressure which should be removed from Eq.(1).
We replaced σ (sigma) with τ as requested. Thanks for noting this very important inaccuracy!

2. Section 5.1, I would rephrase "Symmetrical Multiprocessor (SMP) system with an Intel Xeon E5 processor and 32 GB RAM". "Symmetrical Multiprocessor (SMP)" is very uncommon and adds no further information. Instead, it would be interesting to know how many cores that Xeon E5 cpu has, and how they where (or not) used in the benchmark runs.
We rephrased as requested also added a few more technical details (line 255-257). Also removed a sentence in line 297, which is now superfluous.

Dear editor,

thank you very much for the summary above and for your kind consideration.

We included the changes above and a few more inconsistencies that had previously gone unnoticed.

Contact author:
please note that we replaced the contact author, which is now Paul Bons (paul.bons@uni-tuebingen.de)

Main text:
- FIXED: Line 17: replaced ‚FE' by ‚finite elements' (used for the first time)
- FIXED: Line 35: 'that that' replaced by 'that'
- FIXED: Line 74/75: small changes of the wording due to renaming of deviatoric stress tensor
- FIXED: Due to wrong automatic numbering, Eq. 14 and 15 had been labed 8 and 9 in the previous version. This has been corrected, and mentions in lines 203, 218, 223, 277, 355 were adapted.

Supplement text:
- FIXED: some tables had a formatting issue and didn't show borders.